# Intention to use maternity waiting home and associated factors among pregnant women in Gamo Gofa zone, Southern Ethiopia, 2019

**Wubishet Gezimu[1], Yibelu Bazezew Bitewa** **[2]\*, Mekuanint Taddele Tesema[3], Tewodros Eshete Wonde[3]**

**1** Department of Nursing, College of Health Sciences, Mettu University, Mettu, Ethiopia, **2** Department of Midwifery, College of Health Sciences, Debre Markos University, Debre Markos, Ethiopia, **3** Department of Public Health, College of Health Sciences, Debre Markos University, Debre Markos, Ethiopia

\* yibebazezew5106@gmail.com

**Data Availability Statement:** All relevant data are within the paper and its Supporting information files.

## Abstract

### Background

A maternity waiting home is a temporary residence in which pregnant women from remote areas wait for their childbirth. It is an approach targeted to advance access to emergency obstetric care services especially, in hard-to-reach areas to escalate institutional delivery to reduce complications that occur during childbirth. Apart from the availability of this service, the intention of pregnant women to utilize the existing service is very important to achieve its goals. Thus, this study aimed to assess the intention to use maternity waiting homes and associated factors among pregnant women.

### Methods

Community-based cross-sectional study was conducted among 605 pregnant women using a multistage sampling technique from March 10 to April 10, 2019, by using a structured questionnaire through a face-to-face interview. The collected data was entered into Epi-Data version 3.1 and analyzed using the SPSS version 24 statistical package. Logistic regression analysis was used to test the association. All variables at p-value < 0.25 in bivariate analysis were entered into multivariate analysis. Lastly, a significant association was declared at a P-value of < 0.05 with 95% CI.

### Results

In this study, the intention to use maternity waiting homes was 295(48.8%, 95%CI: 47%-55%)). Occupation (government employee) (AOR:2.87,95%CI: 1.54–5.36), previous childbirth history (AOR:2.1,95%CI:1.22–3.57), past experience in maternity waiting home use AOR:4.35,95%CI:2.63–7.18), direct (AOR:1.57,95%CI:1.01–2.47) and indirect (AOR: 2.18, 1.38,3.44) subject norms and direct (AOR:3.00,95%CI:2.03–4.43), and indirect (AOR = 1.84,95%CI:1.25–2.71) perceived behavioral control of respondents were significantly associated variables with intention to use maternity waiting home.

**Funding:** The author(s) received no specific funding for this work.

**Competing interests:** The authors have declared that no competing interests exist.

**Abbreviations:** **ANC**, Antenatal Care; **AOR**, Adjusted Odds Ratio; **EDHS**, Ethiopian Demographic Health Survey; **EmOC**, Emergency Obstetric Care; **HEW**, Health Extension Worker; **MM**, Maternal Mortality; **MMR**, Maternal Mortality Ratio; **MWH**, Maternal Waiting Home; **NGO**, Non-Governmental Organization; **SBA**, Skilled Birth Attendant; **TPB**, Theory of Planned Behavior; **TRA**, Theory of Reasoned Action.

## Conclusion

The magnitude of intention to use maternity waiting homes among pregnant women is low. Community disapproval, low self-efficacy, maternal employment, history of previous birth, and past experiences of MWHs utilization are predictors of intention to use MWHs, and intervention programs, such as health education, strengthening and integration of community in health system programs need to be provided.

## Introduction

The World Health Organization (WHO) defined Maternity Waiting Home (MWH) as a temporary housing service nearby health facilities, in which pregnant women wait for childbirth [1]. Endorsing MWH near health facilities is a strategy that can reduce inequity, by improving poorer women's access to health facilities that enable advanced management of childbirth complications. It is a highly profitable and cheap approach to decrease maternal morbidity and mortality as well as it is a low-cost solution to access skilled birth attendants in remote areas [1–4]. Besides, it is a life-changing innovative approach, which was established as one of the three institutional innovations (existence of community-based health insurances, the establishment of maternal waiting homes, and strengthening of health facilities and personnel) that have substantially contributed to accelerating progress on maternal health in Africa [5]. Even if there are certain challenges to utilize MWH, it is important to educate, and counsel the women about pregnancy, childbirth, newborn and infant care, and family planning, and it offers the opportunity for women to come closer to the health institutions before labor starts which helps to avoid reluctance to walk a long distance after labor begins [6–10].

Globally, about 10.7 million women died in a year between 1990 and 2015 due to maternal causes. This catastrophic event is coarsely 20 folds higher in developing regions than developed regions. Approximately 99% of the estimated global maternal deaths occurred in developing regions in 2015. Of this sub-Saharan Africa alone accounted for roughly 66% of maternal deaths followed by Southern Asia [11]. According to the United Nations sustainable development goal (SDG) three target one plan, the global maternal mortality ratio will be less than 70 per 100,000 live births by 2030 and thus, each country in the world will be required to reduce MMR by at least 7.5% each year between 2016 and 2030 to achieve this target [12].

Access to comprehensive emergency obstetric care is limited in Ethiopia; and due to lack of modern transport, people use a locally made stretcher to carry laboring mothers to the health facility by community members [13]. In Ethiopia, a maternal mortality ratio remains high; which is 412 per 100,000 live births as per the 2016 EDHS report [14]. According to the mini EDHS 2019 survey report, institutional delivery is only 48%, which indicated that home delivery is still common, and as a result of distance, inaccessibility, and lack of appropriate facilities, lack of access to health facilities in rural areas (primarily in hard-to-reach areas), and in rural Ethiopia, only 43%, and 40% of women delivered by skilled birth attendants, and in health facilities respectively as per the 2019 EDHS report [15]. In the Southern Nations, Nationalities, and Peoples' Region of Ethiopia, more than three fourth (78.6%) of women gave birth without a skilled birth attendant, and this region has a high MMR [14, 16]. Though escalating institutional deliveries is vital to reduce maternal and neonatal mortality and building and using maternal waiting homes is one of the solutions [3, 17, 18]. In Ethiopia, the first MWH was built in 1976 [13], and currently, the Amhara region is the top in MWHs coverage with 72%

followed by Southern Nations, Nationalities, and Peoples' Region(SNNPR) (57%), and Oromia region (56%); and least (8%) in the Gambella region [19].

In the developing world, the probability of death among MWH users is 80% less than in non-users and 73% less occurrence of stillbirth among users, and also 98.8% of MWH users delivered with a low proportion of bad obstetric outcomes than non-users [20–23]. The health-seeking behavior of the mothers, inadequate progressive planning for delivery; the women's perception of service benefits (attitude); women's perception about social pressure from important others (normative beliefs); the previous history of obstetric complications, and the availability of MWH at health institutions are factors that are related to the willingness of the women to utilize MWH [24, 25]. Although the evidence showed MWHs as a strategy to reduce MMR and stillbirth [26], there is limited data on the level of utilization of MWH in Ethiopia and the intention of pregnant women to use this service. So, this study was directed to assess the intention to use MWH and associated factors such as pregnant women's attitude, subjective norm, perceived behavioral control, socio-demographic, and obstetrics related to factors [Fig 1].

## Methods

### Study area and period

This study was conducted in the Gamo Gofa zone, Southern Ethiopia using data of pregnant women of Kamba district from March 10 to April 10, 2019. Kamba district is one of the districts' in the SNNPR of Ethiopia, part of the Gamo Gofa Zone. It is located 615 km far from Addis Ababa, 390, and 110 km from the capital of SNNPR, Hawassa, and Arba Minch, capital of Gamo Gofa Zone respectively. According to the 2007 national census conducted by the Central Statistical Agency of Ethiopia, this district has a total population of about 155979, of whom about 79273 and 76706 are men and women respectively; 4072 or 3.02% are urban inhabitants. And a total of 30,180 households were counted in this district, resulting in an average of 4.4 persons to a household, and 29,565 housing units. Regarding health facility coverage, Kamba district has one district hospital, 7 health centers each with MWHs, 2 Satellite clinics, and 39 Health Posts. The estimated number of pregnant women in the area was 7163. There are 38 rural and 5 urban administrative kebeles (the smallest administrative units of Ethiopia) found in this district [33]. Of these, the study was collected on randomly selected 11 kebeles [S1 Fig].

### Study design and population

A community-based cross-sectional study was conducted using all pregnant women who were living in the Gamo Gofa zone as a source population, and all registered pregnant women in the selected kebeles of Kamba district as the study population. Those pregnant women who lived less than six months in the study area and those who delivered by cesarean section were excluded from the study.

### Sampling and study variables

The assumptions used to calculate the maximum sample size for this study were population proportion of intention to use MWHs, 57.3% which was taken from a study done in Jimma district [34] with a 95% confidence level (CI) and an estimated margin of error 5%. With none-response rate of 10% which yields 38, the minimum required sample size = 38 + 376 = 414. Since the sampling technique was multistage, the design effect of 1.5 was considered, and the final total sample size was 621. This study was conducted in the Gamo Gofa zone

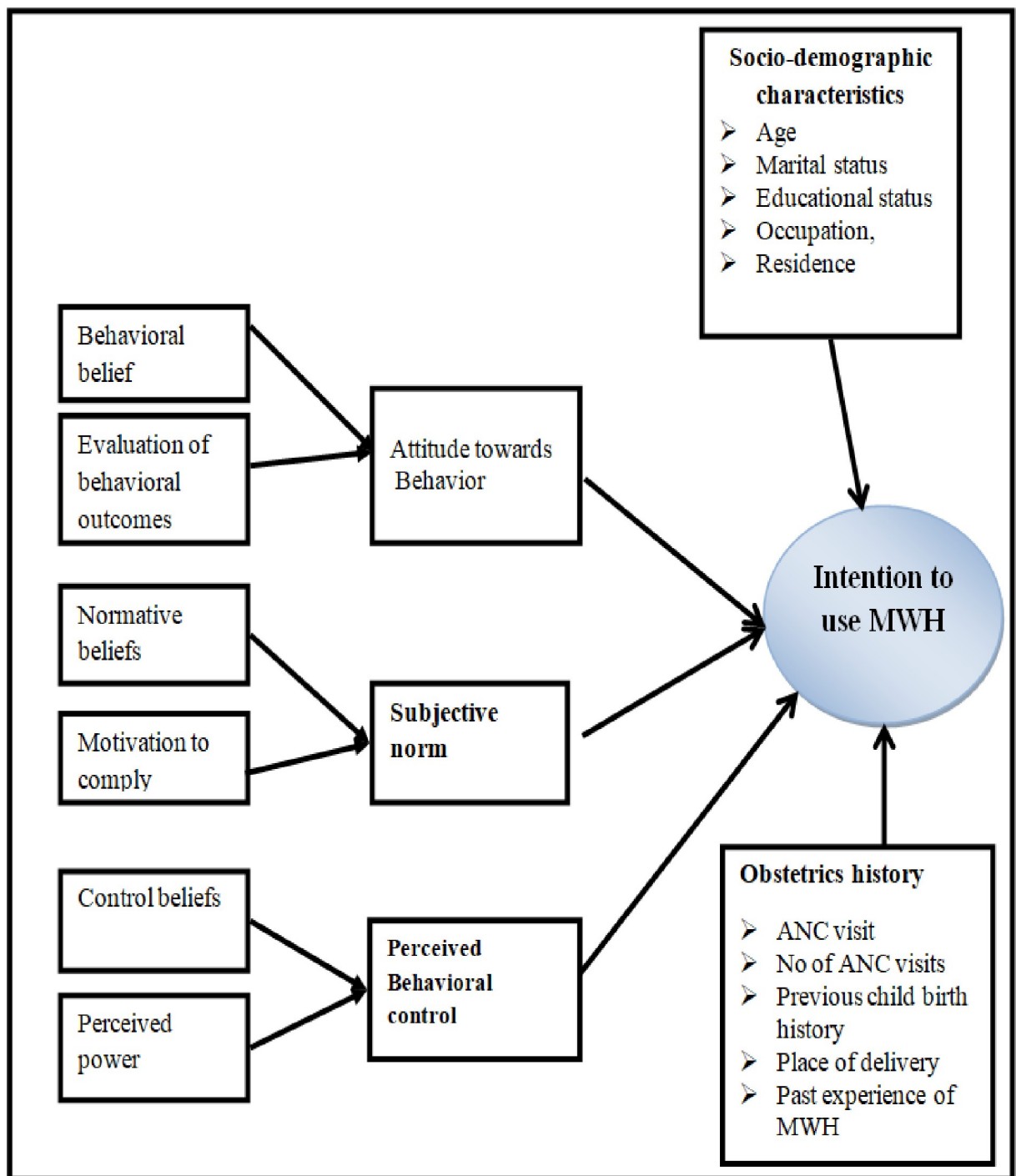

**Fig 1. Conceptual framework, which was adapted from the theory of planned behavior, showed factors associated with intention to use maternity waiting home [27–34].**

in southern Ethiopia using the data of pregnant women in the Kamba district, and to run the multistage sampling technique districts in this zone were considered as clusters because there is a homogeneous feature between districts. Then, due to the number of pregnant women estimation in each district and our sample size, the Kamba district was selected randomly. In the

Kamba district there are 43 kebeles, and to make the sample being representative to all Kebeles, we took 25% of them (11 kebeles) randomly. Additionally, we employed a design effect of 1.5 to represent the sample to the Gamo Gofa Zone. By considering the heterogeneous feature of Kebeles within a cluster, proportional allocation of pregnant women was done in each selected stratum (Kebele), and in each Kebele, there are assigned HEWs, and these HEWs provide community visits at the household level at least 2 times week and one of their responsibility is to investigate whether pregnant woman/s is present in that household or not, and then to register them. The registration includes the gestational age, EDD, Name, Phone number (if available), the exact location, and others. After obtaining the registration number of pregnant women from health extension workers (HEWs) of each selected Kebele, a simple random sampling technique (lottery method) was applied by using the registration book of pregnant women as a framework.

Furthermore, 621 pregnant women who fulfilled the inclusion criteria were recruited based on their registration number via simple random sampling technique, and an interview was done at the household level for those randomly selected pregnant women [S1 Fig], and variables such as **socio-demographic characteristics** (age, marital status, religion, educational status, and occupation), **obstetric history** (current ANC visit, number of ANC visit, previous childbirth history, place of delivery, and experience of MWH), and **psycho-social/behavioral variables** (direct and indirect attitude, direct and indirect subjective norm, and also direct and indirect perceived behavioral control) were included in the data collection tool [Fig 1].

## Operational definitions

**Intention to use Maternity Waiting Home (MWH).** An indication of pregnant women's willingness, and how much effort they are planning, and exert to utilize it. It was measured by questions containing five points Likert scale. Lastly, it was dichotomized into two groups using the mean score. Those who scored above the mean (12.61) were classified as intended to use MWH, whereas those who scored less than or equal to the mean score (12.61) were considered as not intended to use MWH [35].

**Attitude towards MWH.** The degree of pregnant women's maternity waiting home utilization behavior is influenced by her emotions, motivations, perceptions, and thoughts. It was measured by using four questions containing a five-point Likert scale. And they are classified into two by using mean as they have favorable and unfavorable attitudes.

**Favorable attitude.** The respondent's attitude score > mean (11.59), and

**Unfavorable attitude.** The respondent's attitude score less than or equal to the mean score [35].

**Indirect attitude.** It was computed by multiplying behavioral beliefs of pregnant women concerning each outcome by corresponding outcome evaluation ratings, and then summing these product scores across all outcomes. Finally, the mean (15.93) was used to dichotomize into favorable and unfavorable indirect attitudes [32].

**Direct subjective norm.** Pregnant women's perception of using maternity waiting homes. It was measured by four questions containing a five-point Likert scale, and they were classified into two by using mean as they have favorable and unfavorable subjective norms. Those who scored above the mean (10.70) were classified as having favorable subjective norms, but those who scored less than or equal to the mean were classified as having unfavorable subjective norms [35].

**Indirect subjective norm.** It was computed by multiplying pregnant women's normative belief about each referent by her motivation to obey with that referent and then summing

these product scores across all referents. Finally, the mean score (10.46) was used to dichotomize [32].

**Perceived behavioral control.** Each pregnant woman's belief concerning how easy or difficult it is to use maternity waiting homes. It was measured by four questions containing a five-point Likert scale and was classified into two by using mean as they have favorable and unfavorable perceived behavioral control. Those who scored above the mean (11.75) were classified as having favorable perceived behavioral control, and those who scored less than or equal to 11.75 were classified as having unfavorable perceived behavioral control [35].

**Indirect perceived behavioral control.** It was computed by multiplying each control belief of pregnant woman by her corresponding perceived power (impact) scores and then summing these product scores across all control factors, and finally, those who scored above the mean (14.64) were taken as having favorable indirect perceived behavioral control [32].

## Data collection tool, quality control, and procedures

Data were collected using a structured and pre-tested questionnaire through a face-to-face interview. Five trained data collectors (HEWs) who were recruited from health posts, and who were supervised by three supervisors (BSc, Nurses) from health centers were executed for the data collection. The questionnaire was translated from English to Amharic, and then to Gamogna (local language) and back to English by six language experts to assure consistency [S1–S3 Tables]. Face validity was conducted to assess the form of the questionnaire in terms of feasibility, readability, evenness of technique and formatting, and the clearness of the language, that is to assess the presentation and relevance of the measuring tool as to whether the items in the tool emerge to be pertinent, logical, explicit and obvious, and to validate the questionnaire, 2 experts groups on each English, Amharic, and Gamogna language have participated. A pretest was conducted on 5% of the total sample size among pregnant women in one of the non-selected Kebele before the actual data collection. The training was given for data collectors and supervisors on the objective, and purpose of the study, the respondents' rights, and the confidentiality of information, informed consent, and techniques of the interview. Finally, data were entered into Epi data version 3.1, and before conducting any analysis; the data pre-processing tasks like data cleaning, coding and recording, variable re-categorization, and identification for inconsistencies were done.

## Data analysis and process

First, the data were coded and entered into Epi Data version 3.1 and then exported to SPSS version 24 statistical package for further analysis.

Data cleaning was performed to check for frequencies, and missed values then descriptive analysis such as proportions, percentages, means, and measures of dispersion, tables, and graphs were used to describe the data. To test the association between the independent and the outcome variable, logistic regression analysis was done. All variables at a p-value less than 0.05 in bivariate analysis were entered into multivariate analysis to identify the independent association of variables of intention to use MWH. Finally, significant independent associations were declared at a P-value of less than 0.05 with 95%CI. Hosmer-Lemeshow model fitness test was used to indicate the goodness of the final model, and model fitness is assured when the value is insignificant that is greater than 0.05.

## Ethical approval and consent to the participants

Before data collection, Ethical approval was primarily obtained from Debre Markos University research ethical approval committee. Likewise, an official ethical clearance letter was obtained

from the Kamba district health bureau. Following an explanation of the purpose of the study, verbal informed consent was obtained from each participant. Also, confirmation was made that they are free to withdraw and discontinue participation without any form of prejudgments. Confidentiality of information and the privacy of participants' were assured by making their names anonymous.

## Results

### Prevalence of intention to use maternity waiting homes

A total of 605 pregnant women participated in the study, with an overall response rate of 97.4%. In this study, less than half, 295 (48.8%) with 95% CI (47%-55%) of pregnant women were intended to use maternity waiting homes [S1 Fig], and the mean score of intention to use MWH was 12.61(SD± 4.738).

### Socio-demographic characteristics of pregnant women

The mean age of pregnant women who participated in this study was 25.91 with SD ± 5.023. About 33.4% of respondents were found in the age range of 20–24 years. The majority, 562 (92.90%) of respondents were married and 389 (64.3%) of them were housewives. Furthermore, about thirty percent of respondents were completed primary education [Table 1].

### Maternal health services utilization of pregnant women

More than half of respondents had greater than three pregnancy experiences, and the majority (84.79%) of them had given birth before the current pregnancy. Likewise, about 61.60% of the respondents gave their previous childbirth at health institutions [Table 2].

**Table 1. Socio-demographic characteristics of pregnant women.**

| Variables | Categories | Frequencies | Percentages |
|---|---|---|---|
| **Maternal Age in years** | 15–19 | 52 | 8.60 |
| | 20–24 | 202 | 33.40 |
| | 25–29 | 194 | 32.10 |
| | 30–34 | 127 | 21.00 |
| | 35 and above | 30 | 5.00 |
| **Religion** | Orthodox | 191 | 31.60 |
| | Protestant | 335 | 55.40 |
| | Muslim | 79 | 13.10 |
| **Occupation** | Housewife | 389 | 64.30 |
| | Merchant | 145 | 24.00 |
| | Government employee | 71 | 11.70 |
| **Educational status** | Can't to read and write | 177 | 29.30 |
| | Only read and write | 159 | 26.30 |
| | Primary education | 184 | 30.40 |
| | Secondary education and above | 85 | 14.00 |
| **Marital status** | Married | 562 | 92.90 |
| | Single | 7 | 1.20 |
| | Widowed | 32 | 5.30 |
| | Divorced | 4 | 0.70 |

**Table 2. Maternal health services utilization among pregnant women.**

| Variables | Category | Frequency | Percentages |
|---|---|---|---|
| Number of pregnancy | 1–2 | 253 | 41.80 |
| | >3 | 352 | 58.20 |
| ANC visit for current pregnancy | Yes | 388 | 64.10 |
| | No | 217 | 35.90 |
| Number of ANC visit(n = 388) | 1st visit | 64 | 16.50 |
| | 2nd visit | 117 | 30.20 |
| | 3rd visit | 118 | 30.40 |
| | 4th visit | 89 | 22.90 |
| Pervious childbirth history | Yes | 513 | 84.80 |
| | No | 92 | 15.20 |
| Place of delivery (n = 513) | Home | 197 | 38.40 |
| | Health institution | 316 | 61.60 |

## Experience in maternity waiting homes utilization

Less than a quarter of the respondents, 130 (21.50%) had past experiences with MWH utilization. Of those who utilized previously, 72 (55.38%) have stayed in the service center for two weeks and only 9 (6.92%) of them have stayed for more than two weeks [Table 3].

## Direct components of Theory of Planned Behavior (TPB)

Two hundred seventy-three (45.10%) of the pregnant women had a favorable attitude to use MWH with a mean of 11.59 (SD ± 4.350). And also, two hundred forty-nine (41.20%) and 327 (54.00%) of the respondents had favorable subjective norm and perceived behavioral control with a mean of 10.70 (SD ±4.565) and 11.75 (SD ±3.723) respectively [Fig 2].

## Indirect TPB components (indirect attitude, indirect subjective norm, and indirect perceived behavioral control)

Four hundred forty-three (73.2%) of pregnant women had favorable indirect attitudes with a mean score of 15.93 (SD±3.455). Four hundred seventy-six (78.70%) of pregnant women agreed that MWH would help them to get a healthy child. Likewise, 500(82.60%) of them agreed MWH would help them to be happy, and reduce labor fear. Additionally, 352 (58.20%),

**Table 3. Past experiences in MWH service utilization among pregnant women.**

| Variables | Categories | frequencies | Percentages |
|---|---|---|---|
| Previous experiences of MWH | No | 475 | 78.50 |
| | Yes | 130 | 21.50 |
| Reasons for previous utilization(n = 130) | Fear of labor illness | 42 | 32.30 |
| | Fear of death related to delivery | 36 | 27.70 |
| | To get better health care from health professionals | 36 | 27.7 |
| | To get a healthy child | 11 | 8.50 |
| | To get enough rest and free from workload | 5 | 3.80 |
| Duration of stay in MWH previously (n = 130) | <15 days | 49 | 37.70 |
| | Only 15 days | 72 | 55.40 |
| | >15 days | 9 | 6.90 |

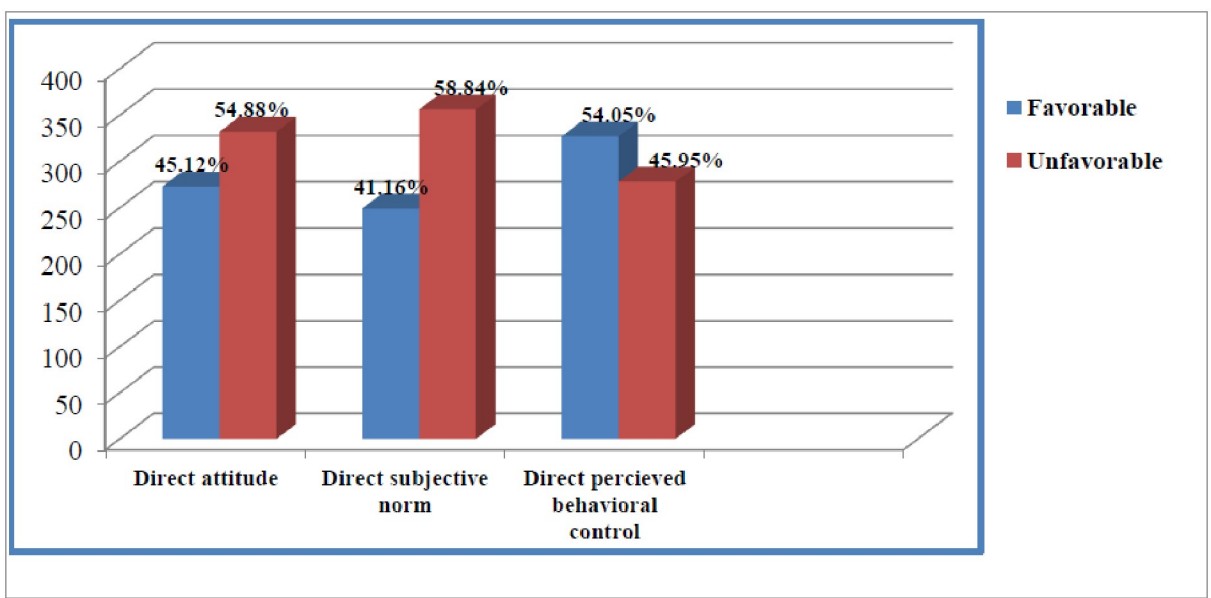

**Fig 2. Proportion of direct components of theory of planned behavior among pregnant women in Gamo Gofa zone, Southern Ethiopia, 2019.**

and three hundred eleven (51.40%) of the respondents mentioned that staying at MWH is very good to get a healthy child and to reduce labor fear respectively.

Two hundred forty-two (40%) of the respondents had favorable indirect subjective norms with a mean score of 10.46 (SD±4.349). Three hundred seven (50.70%) of the participants agreed that HEWs think that they should stay in MWH. Two hundred twelve (35.10%) of them agreed that their husbands think that they should stay in the maternity waiting home. In the motivations to comply with the above normative beliefs, 299 (49.40%) of respondents reported that HEWs' approval for their stay in MWH was much important. Of them, less than a quarter 128 (21.20%), and 111 (18.30%) were reported that their mothers' and husbands' approval to use MWH is much important respectively.

Three hundred nine (51.1%) of the respondents had favorable indirect perceived behavioral control with a mean score of 14.64 (SD±3.738). The majority 495(81.90%) of them reported that it's unlikely to get enough food in MWH to use it, followed by 447 (73.90%) of them reported they were unlikely to get transportation/walk to long distance. And also 379 (62.70%) pregnant women reported it's unlikely to get individuals to take care of their family while they staying in MWH. The power of control belief result revealed that 475 (78.50%) and 447 (73.90%) of the respondents agreed that the availability of food and transport accessibility makes it easy to use MWH respectively.

## Bivariate and multivariate analysis of factors associated with intention to use maternity waiting home

In the bivariate analysis age of the respondents, educational level, occupation, previous childbirth history, the experience of MWH utilization, the reason for past utilization, and duration of stay at MWH, direct and indirect attitude, direct and indirect subjective norm, and direct and indirect perceived behavioral control were associated with intention to use MWH. However, in the multivariate regression analysis occupation (government employees), previous childbirth history, the experience of MWH, direct and indirect subjective norm, and perceived

**Table 4. Bivariate and multivariate analysis of factors associated with intention to use maternity waiting home among pregnant women.**

| Variables | Categories | Intended to use MWH | Not intended to use MWH | COR (95%CI) | AOR(95%CI) |
|---|---|---|---|---|---|
| **Occupation** | Government employees | 49(8.1%) | 22(3.63%) | 2.51(1.46–4.30) | 2.87(1.54–5.36)** |
| | Merchant | 63(10.41%) | 82(13.55%) | 0.86(0.59–1.27) | 0.69(0.44–1.10) |
| | House wife | 183(30.25%) | 206(34.06%) | 1 | 1 |
| Previous history of child birth | Yes | 262(43.30%) | 251(41.48%) | 1.86(1.17–2.95) | 2.08(1.22–3.57)* |
| | No | 33(5.45%) | 59(9.75%) | 1 | 1 |
| Past experience of MWH | Yes | 102(16.86%) | 28(4.63%) | 5.32(3.37–8.40) | 4.35(2.63–7.18)** |
| | No | 193(31.90%) | 282(46.61) | 1 | 1 |
| Direct Subjective Norm | Favorable | 155(25.62%) | 94(15.54%) | 2.54(1.82–3.55) | 1.57(1.01–2.47)* |
| | Unfavorable | 140(23.14%) | 216(35.70%) | 1 | 1 |
| Direct Perceived behavioral Control | Favorable | 208(34.38%) | 119(19.67%) | 3.83(2.73–5.38) | 3.00(2.03–4.43)** |
| | Unfavorable | 87(14.38%) | 191(31.57%) | 1 | 1 |
| Indirect subjective norm | Favorable | 162(26.78%) | 74(12.23%) | 3.88(2.74–5.49) | 2.18(1.38–3.44)** |
| | Unfavorable | 133(21.98%) | 236(39.01%) | 1 | 1 |
| Indirect Perceived behavioral Control | Favorable | 165(27.27%) | 144(23.80%) | 1.46(1.06–2.01) | 1.84(1.25–2.70)* |
| | Unfavorable | 130(21.49%) | 166(27.44%) | 1 | 1 |

** = p <0.001: strongly significant association;

* = p<0.05: statistically significant, and 1 = reference group

behavioral control of the respondents were the factors that significantly associated with intention to use maternity waiting home. Respondents who were government employers were 2.87 times (AOR: 2.87, 95%CI: 1.54–5.36) more likely to intend to use MWH as compared to housewives. Pregnant women who had previous childbirth history were 2.1 times (AOR: 2.08, 95%CI: 1.22–3.57) more likely to intend to use MWH than those pregnant women who had not given birth before. According to the findings of this study, pregnant women who had experience in MWH utilization were 4.35 times (AOR: 4.35, 95%CI: 2.63–7.18) more likely intended to use MWH as compared to those who have not utilized it in the past.

Pregnant women with favorable direct subjective norm were 57% (AOR: 1.57, 95%CI: 1.01–2.47) more likely intended to use MWH as compared to those with the unfavorable subjective norm. This study also revealed that indirect subjective norm has a significant association with intention to use MWM i.e. respondents who have favorable indirect subjective norm were 2.2 times (AOR: 2.18, 95%CI: 1.38–3.44) more likely intended to use MWH as compared to those who have an unfavorable indirect subjective norm. The finding from this study showed that both direct and indirect perceived behavioral control have a significant association with intention to use MWH i.e. pregnant women who had favorable direct, and indirect perceived behavioral control was3 times (AOR: 3.00, 95%CI: 2.03–4.43), and84% (AOR: 1.84, 95%CI: 1.25–2.70) more likely intended to use MWH as compared to those with unfavorable direct and indirect perceived behavioral controls respectively [Table 4].

## Discussion

This study was conducted to identify the intention to use the maternity waiting homes and associated factors among pregnant women in southern Ethiopia. According to this study, 48.8% (95%CI: 47–55%) of pregnant women were intended to use maternity waiting homes, which indicated that more than 50% of the respondents were not willing to use MWHs. And if they are not willing to stay in the maternity waiting homes, especially for those pregnant women who are from remote areas, the probability of delay to reach health institutions and

giving birth at home might be increased which results in the development of different life endanger complication of both the women and their fetuses. This finding is consistent with a previous study conducted in Mettu district, Illubabor zone, Ethiopia which is 48.80% [35]. However, this finding is lower than other previous studies conducted in Jimma district, Ethiopia, which was 57.3% [34] and Butajira, southern Ethiopia, it was 55.1% [36] and a study conducted in Kenya which was 61.1% [28]. These discrepancies could be due to variations in socio-demographic differences among the study areas; disparities in the health service utilization and accessibility; a gap of knowledge about MWH services among the study populations and also the nature of the study. This 48.8% of intention to use maternity waiting homes is greater than a prior study conducted in rural health centers of Ethiopia, which was 27% [29], and a study conducted in rural Kenya which revealed 45% of women intended to use maternity waiting homes [37]. The reasons for this discrepancy might be due to variations in the study period, and the study population.

In this study, the occupation of pregnant women that is being government employees was found to be one of the significantly associated variables with the intention of pregnant women to use MWH as compared to housewives. In contrary to this finding, a study conducted in the Jimma Zone, Ethiopia indicated that housewives had higher odds of the utilization of MWH than farmers/traders/others [38]. Nonetheless, there has been limited scientific evidence concerning this significant association, the possible reason for this association could be due to those women who are government employees might have exposure to information and better insight about MWH services as compared to those women who are housewives.

Intention to use MWH was significantly associated with previous childbirth history. Even if there is no enough data regarding this significant association, the possible reason could be those women who had a history of previous childbirth might have better information about the availability and importance of MWH services during previous health facility visiting than those women who had not ever given birth.

According to the findings of this study, intention to use MWH among pregnant women has a statistically significant relationship with experience in MWH utilization. This finding is supported by a study done in the Jimma district which revealed that women who had the experience of MWH utilization were 16.3% more intended to use MWH than those who do not have experience [34]. The possible reason for this significant association could be; those women who utilized MWH in the past might have better insight into the benefit of staying in MWHs as compared with women who were not ever utilized it.

This study revealed that direct subjective norm was significantly associated with intention to use maternity waiting home which indicated community (most important persons) approval or disapproval to use MWHs affects pregnant women's intention to utilize it. As a result, intention to use MWH was significantly associated with pregnant women's favorable direct subjective norm i.e. high community approval to stay in maternity waiting homes increases the intention of pregnant women to wait in it for two weeks before labor starts. This is congruent with a study done in Mettu districts, Ethiopia which depicted that women who had favorable subjective norms were markedly intended to use MWH than their counterparts, and also it is in line with a study conducted in the Jimma district [34, 35]. This could be due to the crucial importance of significant others (husbands, mothers, HEWs, neighbors, and other important persons) of pregnant women in the decision-making process regardless of their willingness to use MWHs.

This study also revealed that indirect subjective norm, which is pregnant women's agreement or disagreement regarding their husband's, mother's, neighbors, and HEWs thinking towards their stay in MWHs, has a noteworthy association with intention to use MWH i.e. pregnant women who have favorable indirect subjective norm (who agreed that other

important persons think they should wait in MHWs) were significantly intended to use MWH, and it is supported with a study done in Jimma district, Ethiopia [34]. In the current study, 50.75% and 35.05%) of the respondents agreed that HEWs and their husbands think that they should stay in MWH. This finding is congruent with a study conducted in Jimma district, Mettu District, and rural Zambia in which HEWs and husbands have participated in the decision to use MWHs [25, 34, 35]. The reason for this association could be due to the valuable significance of HEWs and husbands' role in the utilization of MWH.

The finding from this study showed that direct perceived behavioral control (self-efficacy) that is easiness or difficult to use MWHs has a significant association with intention to use MWH, which indicated an intention to use MWH among pregnant women has an incredible relationship with favorable direct perceived behavioral control of pregnant women i.e. respondents who have low self-efficacy were not willing to stay in maternity waiting homes. This result is compatible with a finding from the Mettu district in which participants with favorable perceived behavioral control were 99% more likely intended to use MWH as compared to unfavorable perceived behavioral control [35], and also it is congruent with a survey done in Jimma, Ethiopia which revealed that direct perceived behavioral control had a significant association with intention to use MWH [34].

In this study, pregnant women who have favorable indirect perceived behavioral control were significantly intended to use MWH as compared to their counterparts which revealed pregnant women's perception on availability of transportation during advanced gestation, long-distance, availability of sufficient food in the waiting homes, accessibility of individuals who give care for their families at the households, and on desolate surrounding in the maternity waiting homes was related on the women's intention to use MWHs. This finding is also congruent with a survey done in Jimma which showed a strong significant association between indirect perceived behavioral control and intention to use MWH [34]. The potential rationale for this evenness could be due to parallel perceived barriers such as long-distance/lack of transportation, unavailability of food in MWHs among the two populations since both of them were conducted among rural districts. This study also revealed 81.82% of the respondents reported that it's difficult to get enough food in MWH during service utilization; it followed by 72.24% of them reported they were unlikely to get transportation or walk to the long-distance, which is supported by previous studies conducted on maternity waiting facilities for improving maternal and neonatal outcome, and in strengthening referral systems in low-resource countries [39, 40]. However, this finding is greater than the study conducted in Butajira in which 41.8% and 33.4% of participants were unlikely to get enough food and transport to go to MWHs respectively [36]. The differences could be due to time variation, socio-demographic differences, and availability of roads and transportation between the populations.

Different kinds of the literature showed the presence of a statically significant association between intention to use MWH and educational level, direct and indirect attitude of pregnant women [34, 35]. However, these variables were not significantly associated with pregnant women's intention to use MWHs in the multivariate analysis of this study despite the presence of association in bivariate analysis. This illustrates intention to use MWHs is not affected as a result of maternal educational level, and by the view of pregnant women about advantages and disadvantages of staying in maternity waiting homes but it is affected by other enabling factors and perceived barriers.

The community-based nature of this study is its strength, however, as this is a quantitative study, it might lack some hidden attitudes of women that can predict the intention and which would be addressed by qualitative type of studies and it is tough to declare a cause-effect relationship between the outcome and independent variables due to the cross-sectional nature of this study. To avoid selection bias, we used a simple random sampling technique to select the

study participants from the list of pregnant women in the health extension workers registration book, and the data were collected in the community at home. However, using the list of pregnant women in the HEWs registration book as a sampling frame may not include new pregnant women and this is one of the limitations of this study. During the interview, social desirability bias might be committed, but to reduce it only volunteer participants were involved, and their names were kept anonymous, a brief overview of the study was provided, we used experienced interviewers, and the interview was carried out using a one-on-one strategy. To avoid interviewer bias, we instruct our interviewers to read each question accurately to the respondents to answer based on their best understanding of the question, do not interpret the question for the respondents, and offer to say again the question precisely as it appears. Generally, we educate our data collectors to avoid any adjustment, explanation, addition, subtraction, suggestion, or change in verbal variation during the interview process. But using HEWs to collect the data is also the other limitation of this study which imposes social desirability bias.

## Conclusions

Generally, the intention to use maternity waiting homes among pregnant women is low, which leads to inaccessibility of obstetric care and it will provoke pregnant women to give birth at home without skilled birth attendants that may end up with the development of different life-threatening complications including maternal and neonatal deaths. Community disapproval, low self-efficacy, maternal employment, history of previous childbirth, and past experiences of MWHs utilization are predictors of intention to use MWHs. These findings call for the intervention of the community to increase the intention of pregnant women towards MWH utilization and we authors recommend that:

1. It is better to mobilize the community and provide health education, concerning MWH service to all the community and mothers to improve the subjective norm.

2. Community-based stakeholders such as health development armies need to be strengthened and extended to alleviate transportation problems and food shortages, those upshot women with unfavorable perceived control beliefs, through integration with local and federal governments to upsurge perception of mothers towards behavioral controls.

3. Health care professionals need to advise and counsel the mothers on MWH service at ANC visiting to increase their intention.

4. In this area, it is better to do further researches in a qualitative approach.

## Supporting information

**S1 Fig. Diagrammatic presentation of sampling procedure used to recruit pregnant women in Kamba District, Gamo Gofa Zone, Southern Ethiopia, 2019.**
(DOCX)

**S2 Fig. Magnitude of intention to use maternity waiting home among pregnant women in Gamo Gofa zone, Southern Ethiopia, 2019.**
(DOCX)

**S1 Table. English version questionnaire.**
(DOCX)

**S2 Table. Amharic version questionnaire.**
(DOCX)

**S3 Table. Gamogna version questionnaire.**
(DOCX)

**S1 File. English version consent form.**
(DOCX)

**S2 File. Amharic version consent form.**
(DOCX)

**S3 File. Gamogna version consent form.**
(DOCX)

**S1 Data.**
(XLSX)

## Acknowledgments

The authors would like to thank data collectors and supervisors for their assiduousness and dedication in collecting and inputting high-quality data and also we would like to thank the study participants for their willingness to participate kindly provision of the necessary information, and sacrifice of their valuable time.

## Author Contributions

**Conceptualization:** Wubishet Gezimu, Yibelu Bazezew Bitewa.

**Data curation:** Wubishet Gezimu, Yibelu Bazezew Bitewa.

**Formal analysis:** Wubishet Gezimu, Yibelu Bazezew Bitewa, Tewodros Eshete Wonde.

**Investigation:** Wubishet Gezimu, Yibelu Bazezew Bitewa, Mekuanint Taddele Tesema, Tewodros Eshete Wonde.

**Methodology:** Wubishet Gezimu, Yibelu Bazezew Bitewa, Mekuanint Taddele Tesema, Tewodros Eshete Wonde.

**Software:** Yibelu Bazezew Bitewa, Tewodros Eshete Wonde.

**Supervision:** Yibelu Bazezew Bitewa, Mekuanint Taddele Tesema.

**Validation:** Wubishet Gezimu, Yibelu Bazezew Bitewa.

**Writing – original draft:** Wubishet Gezimu, Yibelu Bazezew Bitewa, Tewodros Eshete Wonde.

**Writing – review & editing:** Wubishet Gezimu, Yibelu Bazezew Bitewa, Mekuanint Taddele Tesema, Tewodros Eshete Wonde.

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
