## [Decision Letter · Decision Letter 0]

11 Nov 2020

PONE-D-20-31704

Intention to use maternity waiting home and associated factors among pregnant women in Kamba District, Gamo Gofa zone, Southern Ethiopia, 2019

PLOS ONE

Dear Dr. Bitewa,

Thank you for submitting your manuscript to PLOS ONE. After careful consideration, we feel that it has merit but does not fully meet PLOS ONE’s publication criteria as it currently stands. Therefore, we invite you to submit a revised version of the manuscript that addresses the points raised during the review process.

Two experts in the field handled your manuscript, and we are very thankful for their time and efforts. Although some interest was found in your study, several comments arose that need to be addressed. Please respond to ALL of the reviewers' comments in your revised manuscript.

We look forward to receiving your revised manuscript.

Kind regards,

Frank T. Spradley

Academic Editor

PLOS ONE

2. In your Methods section, please provide additional information about the participant recruitment method and the demographic details of your participants. Please ensure you have provided sufficient details to replicate the analyses such as:   

-    a description of any inclusion/exclusion criteria that were applied to participant recruitment,

-    a table of relevant demographic details,

-    a statement as to whether your sample can be considered representative of a larger population,

-    a description of how participants were recruited, and

-       descriptions of where participants were recruited and where the research took place.

3. Please include additional information regarding the survey or questionnaire used in the study and ensure that you have provided sufficient details that others could replicate the analyses. For instance, if you developed a questionnaire as part of this study and it is not under a copyright more restrictive than CC-BY, please include a copy, in both the original language and English, as Supporting Information.  If the original language is written in non-Latin characters, for example Amharic, Chinese, or Korean, please use a file format that ensures these characters are visible.

4. Please state whether you validated the questionnaire prior to testing on study participants. Please provide details regarding the validation group within the methods section.

5. Please provide additional details regarding participant consent. In the ethics statement in the Methods and online submission information, please ensure that you have specified what type you obtained (for instance, written or verbal, and if verbal, how it was documented and witnessed). If your study included minors, state whether you obtained consent from parents or guardians. If the need for consent was waived by the ethics committee, please include this information.

7. Please include your tables as part of your main manuscript and remove the individual files. Please note that supplementary tables should  be uploaded as separate "supporting information" files.

Reviewers' comments:

Reviewer's Responses to Questions

**Comments to the Author**

1. Is the manuscript technically sound, and do the data support the conclusions?

Reviewer #1: Yes

Reviewer #2: Yes

2. Has the statistical analysis been performed appropriately and rigorously? 

Reviewer #1: Yes

Reviewer #2: Yes

3. Have the authors made all data underlying the findings in their manuscript fully available?

Reviewer #1: Yes

Reviewer #2: No

4. Is the manuscript presented in an intelligible fashion and written in standard English?

Reviewer #1: No

Reviewer #2: No

5. Review Comments to the Author

Reviewer #1: General comments

1. General comments

1.1: Authors should read thoroughly the author’s submission guidelines and strictly follow it for all sections of the manuscript. For this purpose they should download the sample pdf of abstract and sample pdf of manuscript. They should strictly follow the guideline for font size, writing style – upper and lower case writing in title, affiliation, abstract, main manuscript, table titles, figure titles, and references.

1.2: While submitting manuscript authors did not make the manuscript in double space, page numbers and line numbers.

2. English language has to be improved.

2. Title

2.1: Page (P) 2, lines (L) – 7, 9, While writing affiliation, instead of writing College of Health sciences, it should be College of Health Sciences.

2.2: P2, L10 – it should be Department of Public Health, College of Health Sciences

3. Abstract

3.1: P4, L56 – research aim is not the section in abstract. Merge it in background.

3.2: P4, L59 – delete add comma after April 10, and delete /

3.3: P4, L60 – it should be questionnaire and interview.

3.4: P4, L66 – it should be 95%.

3.5: P4, L68 – it should be (AOR

3.6: P 4, L72 – Recommendation is not there – follow guidelines

3.7: P4, L77 – Delete Kamba district and write Ethiopia. For writing Keywords – they should match with MeSH terms.

4. Background

4.1: P5, L79 – It should be World Health Organization

4.2: P5, L81 – It should be health facility instead of hospitals. In your study area there are primary health care units. They are not hospitals.

4.3: P5, L93 – It should be long distance instead of several distances.

4.4: P5, L98 – it should be population instead of populations.

4.5: P5, L101 – give long form of SNNPR as it is appearing first time. It should be Oromia instead of Oromo region. Add and after (56%); and least

4.6: P5, L133 – Add per – it should be as --- attendants as per the 2016.

Comment: Authors did not cite the following publications to add in the background or discussion

a. Kurji J, Gebretsadik LA, Wordofa MA, Sudhakar M, Asefa Y, Kiros G, Mamo A, Bergen N, Asfaw S, Bedru KH, Bulcha G, Labonte R, Taljaard M, Kulkarni M. Factors associated with maternity waiting home use among women in Jimma Zone, Ethiopia: a multilevel cross-sectional analysis. BMJ Open. 2019 Aug 28;9(8):e028210. doi: 10.1136/bmjopen-2018-028210. PMID: 31467047; PMCID: PMC6720516.

b. van Lonkhuijzen L, Stekelenburg J, van Roosmalen J. Maternity waiting facilities for improving maternal and neonatal outcome in low-resource countries. Cochrane Database Syst Rev. 2012 Oct 17;10(10):CD006759. doi:10.1002/14651858.CD006759.pub3. PMID: 23076927; PMCID: PMC4098659.

c. Swanson DL, Franklin HL, Swanson JO, Goldenberg RL, McClure EM, Mirza W, Muyodi D, Figueroa L, Goldsmith N, Kanaiza N, Naqvi F, Pineda IS, López-Gomez W, Hamsumonde D, Bolamba VL, Newman JE, Fogleman EV, Saleem S, Esamai F, Bucher S, Liechty EA, Garces AL, Krebs NF, Hambidge KM, Chomba E, Bauserman M, Mwenechanya M, Carlo WA, Tshefu A, Lokangaka A, Bose CL, Nathan RO. Including ultrasound scans in antenatal care in low-resource settings: Considering the complementarity of obstetric ultrasound screening and maternity waiting homes in strengthening referral systems in low-resource, rural settings. Semin Perinatol.

2019 Aug;43(5):273-281. doi: 10.1053/j.semperi.2019.03.017. Epub 2019 Mar 16. PMID: 30979599; PMCID: PMC6597951.

d. Buser JM, Lori JR. Newborn Outcomes and Maternity Waiting Homes in Low and Middle-Income Countries: A Scoping Review. Matern Child Health J. 2017 Apr;21(4):760-769. doi: 10.1007/s10995-016-2162-2. PMID: 27475822.

e. Gaym A, Pearson L, Soe K. Maternity waiting homes in Ethiopia – three decades experience…..

5. Methods

Comment:

a. Follow guideline for section and sub-sections.

5.1: P8, L – 170, make it as April 10, and delete /

5.2: P8, L – 171, Authors should decide whether it should be as Gamo Gofa Zone or it should be used as Gamo Gofa zone in the whole manuscript including titles of tables and figures. Also authors should decide whether the percentages should be presented with one digit or two digit e.g. 3.2 or 3.21. These percentages be consist in the whole manuscript including tables and figures.

5.3: P8, L – 174 and 175, Instead of actual figures of population it can be written as about 160,000 and % male and female.

5.4: P8, L – 190 – 194 either delete this or put it in supplementary information. This formula is not required.

5.5: P9, L – 198, Explain what is Kebele – International reader will not understand.

5.6: P9, L – 203, Fig.2 - This figure can go into supplementary information.

5.6: P9 – There is need to link these two paragraphs giving information about data collection details. (L – 203 and L – 204).

5.7: P9 – Line – 208. Write as insert Fig. 1 here.

5.8: P10 – Line – 225. Write value of mean in the bracket. It should read – the mean () was to ----------

5.9: P10 – Line – 234. Write value of mean in the bracket. It should read – the mean () was to ----------

5.10: P10 – L 237 – Instead of scale and they – it should be scale and were

5.11: P10 – L 247 – 248 – it should be --- health posts who were supervised by three supervisors (B.Sc, Nurses) from health centre.

5.12: P11 – L 270 – Why this statement is required. Earlier you have not described this model. Or describe it little bit it here.

6. Results

Comment: Follow author’s guidelines for paragraph and sub-paragraphs font size, fond type, etc. Also decide number of decimals one or two to be presented percentages in the manuscript and Tables, and figures. It should be consistent.

6.1: P12 - L – 286 – Type – Insert table 1 here

6.2: P12 - L – 290 - Type – Insert table 2 here

6.3: P12 - L – 294 - Type – Insert table 3 here

6.4: P12 - L – 297 - Type – Insert Figure 3 here

6.5: P12 - L – 303 - Type – Insert Figure 4 here

Comment: Your result section description is short. In bivariate and multivariate section you bring AOR descriptions here from discussion section. In discussions do not put analysis figures. Give probable reasons, compare with other studies.

7. Discussions

Comment: Discussion is quite long.

7.1: P14 L – 338 – Instead of Intention it should be intention.

7.2: P15 L – 373 – 374 – It should be --- women who did not use it.

7.3: P16 L – 391 – 392 – It should be --- (16,25,26).

7.4: P16 L – 397 – it should be – perceived

7.5: P16 L – 414 – whether it is unlikely or likely. Check again.

7.6: P16 L – 415 – it should be – -----socio-demographic differences and availability of road and transportation between the populations.

8. Recommendation

Comment: Follow the guidelines of journal. No separate recommendation section. It should be included in conclusions.

9. Declaration

9.1: P18 – L 451 – it should be sacrifice

9.2: P18 – L 458 – Debre Markos University collage of health sciences

10. References

Comment: Follow guidelines for references – how to write authors, and other details of article of the journal, book, chapter in book, website.

10.1: P19 – L 473 – check the authors list……

10.2: P19 – L 485 – Check authors

10.3: P19 – L 488 – check the authors

10.4: P19 – L 494 – check the author, this is not complete.

10.5: P19 – L 499 – why 2012.

10.6: P20 – L 502 – it should be – comma after Columbia University,

10.7: P20 – L 506 – 509 – No. of authors are more than 6. As Vancoure system cannot be more than 6 authors --- after 6th author it should be et.al.

10.8: P20 – L 510 – What is Ruiter?

10.9: P21 – L 535 – 536 – Is this master’s thesis? Which University? Title of this study and this reference is same.

10.10: P21 – L 543 - what is this xxx – xxx – xxx.

10.11: P21 – L 545 – name of journal, vol. and page numbers

10.12: P21 – L 548 – no. of authors 6 and then et.al

10.13: P21 – L 553 – 555 – Check with reference guide. How to refer?

10.14: P21 – 556 – Is it a chapter in the book?

10.15: P21 – 557 – 558 – check how to refer chapter in book.

11. Tables

11.1: Table 1 – titles should be as per guidelines

11.2: Table 2 – titles should be as per guidelines. ANC visit for current pregnancy – yes no. is 386, but next variable – number of ANC visit – total number of 64, 117, 118, 89 is 388. And % are correct with 388 and not with 386.

11.3: Table 3 - titles should be as per guidelines. Delete all % sign. Why one digit percentage is presented for variable – Reasons for previous utilization ---

11.4: Table 4 – titles should be as per guidelines. Below the “*” P=< statistically associated --- should be statistically significant.

12. Figures

12.1: Figure 1: titles should be as per guidelines.

12.2: Figure 2: titles should be as per guidelines. This can go as supplementary attachment.

12.3: Figure 3: titles should be as per guidelines.

12.4: Figure 4: titles should be as per guidelines.

Reviewer #2: It is my pleasure to be designed as a reviewer of this paper, thank you very much!

This a good paper that examined the intention to use MWH, one of the strategies to bridge geographic barriers to access obstetric care.

Comments

Background

1) the background section needs to be restructured for smooth flow from general problem to specific problem statement. In this case, please try to develop themes for each paragraph and link them logically to the issue. You may follow what is intention to use MWHs, what determines women’s intention to use MWHs (individual, community, health system factors) or what is known in the area, what is unknown, contribution of this paper to the scientific knowledge. You need also explain your theoretical framework here. Figure 1 is not described in the text.

2) You use figures from 2016 EDHS data. Please update by 2019 mini-EDHS

3) Page 5, paragraph 3 states transport challenges citing more than 8 years old data. Please update it. The extent problem regarding transport access is not the same as the problem before 2012. Over the last few years, the GoE distributed ambulances to districts to expand access to emergency transport though efficient use and universal access is still a challenge.

Methods

1. the study utilized pregnant women registered in the selected Kebeles as sampling frame and randomly select respondents from that frame. However, it is not well described how pregnant women were registered in the Kebele? How complete the registration was? And how the women or her household was located for interview? It is known the Ethiopian community health information system/family folder or pregnant women registration by HEWs/WDAs are incomplete. As such, this would induce selection bias due to incomplete sampling frame, potentially excluding eligible respondents

2. The analysis section doesn’t mention of account for clustering or not.

3. The outcome variable originally collected as ordinal but later reduced to binary for logistic regression analysis. I think this technique wastes information and may dilute the statistical power. Why you don’t use ordinal regression?

4. Ethical clearance: Not clear whether written or verbal consent was obtained. And was ethical clearance or support letter this study sought from district health office?

Results

1) Figure 2 and 3 are already narrated in the text. No need to present same information with different format.

2) Overall, the result section is presented multiple subheadings without adequate description of the findings. For instance, the Table 4 is narrated in the discussion section. I would bring the results described in the discussion section (narrations with figures with odds ratios) to here.

3) Contrary to other studies, maternal education is not showing effect on the women’s intention to use MWHs. I think it can be related with the model fit or uncontrolled confounding. For instance, being government employee is associated with intention to use MWHs. This could be due to interaction or confounding with education. Please review it again

Discussion

1. Summary the main findings in the first paragraph and dedicate subsequent paragraphs for discussing intention to use and major determinants. Don’t repeat results here (as described above please move the descriptions regarding Table 4 to results section)

2. There are potential biases in this study including selection bias described above, interviewer bias, and social desirability bias that are not mentioned in the discussion section.

3. Conclusion is simply repeating the main finding. Please rewrite to reflect the relevance of the findings to the program, implications to future research, or your concluding comments/take-home messages

4. Some of the recommendations are not grounded from the study findings. For instance, bullet # 2 on page 16.

Language

Finally, there are multiple grammatical errors that needs to edited by someone who has experiences in academic edition. For instance, page 4, paragraph 1 that reads..."one of the three..." not clear what are these; page 5, last paragraph first sentence talks delivery by SBA, but second sentence talks about home delivery. These are not parallel. The third sentence, "...within three.." need to be deleted; page 7, last paragraph first sentence,"... population proportion of 57.3%..." replace by population proportion of intention to use MWHs, 57.3%;

6. PLOS authors have the option to publish the peer review history of their article (what does this mean?). If published, this will include your full peer review and any attached files.

Reviewer #1: No

Reviewer #2: **Yes: **Gizachew Tadele Tiruneh

---

## [Author Response · Author response to Decision Letter 0]

21 Jan 2021

Authors’ response to the Academic Editor comments’ 

Authors’ response: The whole document revised based on PLOS ONE’s publication criteria 

2. In your Methods section, please provide additional information about the participant recruitment method and the demographic details of your participants. Please ensure you have provided sufficient details to replicate the analyses such as: 

- a description of any inclusion/exclusion criteria that were applied to participant recruitment,

Authors’ response: Those pregnant women who lived less than six months in the study area, and those who delivered by caesarean section were excluded from the study. 

- a table of relevant demographic details,

Authors’ response: The relevant demographic details are included under the study area of this study in a text form. 

- a statement as to whether your sample can be considered representative of a larger population,

- description of how participants were recruited, and descriptions of where participants were recruited and where the research took place.

Authors’ response: This study was conducted in Gamo Gofa zone in southern Ethiopia using the data of pregnant woman in Kemba district, and to run the multistage sampling technique districts in this zone were considered as clusters because there is a homogeneous feature between districts. Then, due to the number of pregnant women estimation in each district and our sample size, Kebma district was selected randomly. In Kebma district there are 43 kebeles, and to make the sample being representative to all Kebeles, we took 25% of them (11 kebeles) randomly. Additionally, we employed design effect of 1.5 to represent the sample to Gamo Gofa zone. By considering the heterogeneous feature of Kebeles within a cluster, proportional allocation of pregnant women was done in each selected stratum (Kebele) ,and in each Kebele there are assigned HEWs ,and these HEWs provide community visits at house hold level at least 2 times a week and one of their responsibility is to investigate whether pregnant woman/s is present in that hose hold or not , and then to register them. The registration includes the gestational age, EDD, Name, Phone number (if available), the exact location and others. After obtaining the registration number of pregnant women from health extension workers (HEWs) of each selected Kebele, a simple random sampling technique (lottery method) was applied by using the registration book of pregnant women as a framework. Furthermore, 621 pregnant women who fulfilled the inclusion criteria were recruited based on their registration number via simple random sampling technique, and interview was done at the household level for those randomly selected pregnant women. 

3. Please include additional information regarding the survey or questionnaire used in the study and ensure that you have provided sufficient details that others could replicate the analyses. For instance, if you developed a questionnaire as part of this study and it is not under a copyright more restrictive than CC-BY, please include a copy, in both the original language and English, as Supporting Information. If the original language is written in non-Latin characters, for example Amharic, Chinese, or Korean, please use a file format that ensures these characters are visible.

Authors’ response: The detail of the questionnaire used in this study is included in the supporting information via three languages. 

4. Please state whether you validated the questionnaire prior to testing on study participants. Please provide details regarding the validation group within the methods section.

Authors’ response: Face validity was conducted to assess the form of the questionnaire in terms of feasibility, readability, evenness of techniques and formatting, and the clearness of the language, that is to assess the presentation and relevance of the measuring tool as to whether the items in the tool emerge to be pertinent, logical, explicit and obvious and to validate the questionnaire, 2 experts groups on each English, Amharic, and Gamogna language were participated. 

 5. Please provide additional details regarding participant consent. In the ethics statement in the Methods and online submission information, please ensure that you have specified what type you obtained (for instance, written or verbal, and if verbal, how it was documented and witnessed). If your study included minors, state whether you obtained consent from parents or guardians. If the need for consent was waived by the ethics committee, please include this information.

 Authors’ response: We obtained verbal consent from all participants, and it was documented using the following consent form.

 Consent form 

Hello my name is _________________ (name of data collector). I am____________________ (the data collector briefly introduce him/herself), and I am here to collect data on “Intention to use maternity waiting home and associated factors among pregnant women” for research purpose. The objective of this study is to assess intention to use maternity waiting home and associated factors among pregnant women in Gamo Gofa zone. The benefit of your participation in this study is to improve maternal and neonatal health in your community as well as a country as a whole by increasing institutional delivery. Hence, your trustworthy and frank participation is ultimately important to achieve this goal. All the information that you provide must be kept confidentially, and your name and information will not be disclosed. The information you give is only disclosed to the investigators and they will use it only for this research purposes. You’ve a full right to not respond to all or part of the questions. 

Are you voluntary to participate in this study? 1. Yes 2. No Thank you!!

The name of Data collector: ____________________

Cell phone of data collector: 

Date of data collection: 

Authors’ response: The ethics statement which was appear in declaration section is deleted 

 7. Please include your tables as part of your main manuscript and remove the individual files. Please note that supplementary tables should be uploaded as separate "supporting information" files.

 Authors’ response: Revision is made based on your comments 

Response to Reviewers’

Reviewer #1: General comments

1. General comments

1.1: Authors should read thoroughly the author’s submission guidelines and strictly follow it for all sections of the manuscript. For this purpose they should download the sample pdf of abstract and sample pdf of manuscript. They should strictly follow the guideline for font size, writing style – upper and lower case writing in title, affiliation, abstract, main manuscript, table titles, figure titles, and references.

Authors’ response: We revised the whole document based on the PLOS ONE’S publication criteria 

1.2: While submitting manuscript authors did not make the manuscript in double space, page numbers and line numbers.

Authors’ response: We made a revision, and we include page numbers and line numbers. 

2. English language has to be improved.

Authors’ response: We tried to correct all grammatical, spelling, punctuation, and preposition errors in the whole manuscript with language expert.

2. Title

2.1: Page (P) 2, lines (L) – 7, 9, While writing affiliation, instead of writing College of Health sciences, it should be College of Health Sciences.

Authors’ response: we corrected it as College of Health Sciences

2.2: P2, L10 – it should be Department of Public Health, College of Health Sciences

Authors’ response: we corrected it as Department of Public Health, College of Health Sciences 

3. Abstract

3.1: P4, L56 – research aim is not the section in abstract. Merge it in background.

Authors’ response: Research aim is merged in the background

3.2: P4, L59 – delete add comma after April 10, and delete /

Authors’ response: It is corrected as from March 10 to April 10, 2019

3.3: P4, L60 – it should be questionnaire and interview.

Authors’ response: It is corrected as a questionnaire and interview.

3.4: P4, L66 – it should be 95%.

Authors’ response: It is corrected as 95%

3.5: P4, L68 – it should be (AOR

Authors’ response: AOR is included 

3.6: P 4, L72 – Recommendation is not there – follow guidelines

Authors’ response: Recommendation is deleted 

3.7: P4, L77 – Delete Kemba district and write Ethiopia. For writing Keywords – they should match with MeSH terms. 

Authors’ response: Kemba district is deleted, and the word Ethiopia is included 

4. Background

4.1: P5, L79 – It should be World Health Organization

Authors’ response: It is corrected as World Health Organization

4.2: P5, L81 – It should be health facility instead of hospitals. In your study area there are primary health care units. They are not hospitals.

Authors’ response: It is corrected as health facilities

4.3: P5, L93 – It should be long distance instead of several distances.

Authors’ response: It is corrected as long distance

4.4: P5, L98 – it should be population instead of populations.

Authors’ response: It is corrected as population 

4.5: P5, L101 – give long form of SNNPR as it is appearing first time. It should be Oromia instead of Oromo region. Add and after (56%); and least

Authors’ response: It is corrected as Southern Nations, Nationalities, and Peoples' Region (SNNPR) (57%) and Oromia region (56%); and least (8%) in the Gambella region

4.6: P5, L133 – Add per – it should be as --- attendants as per the 2016.

Authors’ response: the preposition per is added 

Comment: Authors did not cite the following publications to add in the background or discussion

a. Kurji J, Gebretsadik LA, Wordofa MA, Sudhakar M, Asefa Y, Kiros G, Mamo A, Bergen N, Asfaw S, Bedru KH, Bulcha G, Labonte R, Taljaard M, Kulkarni M. Factors associated with maternity waiting home use among women in Jimma Zone, Ethiopia: a multilevel cross-sectional analysis. BMJ Open. 2019 Aug 28;9(8):e028210. doi: 10.1136/bmjopen-2018-028210. PMID: 31467047; PMCID: PMC6720516.

b. van Lonkhuijzen L, Stekelenburg J, van Roosmalen J. Maternity waiting facilities for improving maternal and neonatal outcome in low-resource countries. Cochrane Database Syst Rev. 2012 Oct 17;10(10):CD006759. doi:10.1002/14651858.CD006759.pub3. PMID: 23076927; PMCID: PMC4098659.

c. Swanson DL, Franklin HL, Swanson JO, Goldenberg RL, McClure EM, Mirza W, Muyodi D, Figueroa L, Goldsmith N, Kanaiza N, Naqvi F, Pineda IS, López-Gomez W, Hamsumonde D, Bolamba VL, Newman JE, Fogleman EV, Saleem S, Esamai F, Bucher S, Liechty EA, Garces AL, Krebs NF, Hambidge KM, Chomba E, Bauserman M, Mwenechanya M, Carlo WA, Tshefu A, Lokangaka A, Bose CL, Nathan RO. Including ultrasound scans in antenatal care in low-resource settings: Considering the complementarity of obstetric ultrasound screening and maternity waiting homes in strengthening referral systems in low-resource, rural settings. Semin Perinatol.

2019 Aug;43(5):273-281. doi: 10.1053/j.semperi.2019.03.017. Epub 2019 Mar 16. PMID: 30979599; PMCID: PMC6597951.

d. Buser JM, Lori JR. Newborn Outcomes and Maternity Waiting Homes in Low and Middle-Income Countries: A Scoping Review. Matern Child Health J. 2017 Apr;21(4):760-769. doi: 10.1007/s10995-016-2162-2. PMID: 27475822.

e. Gaym A, Pearson L, Soe K. Maternity waiting homes in Ethiopia – three decades experience…..

Authors’ response: The listed citations included in the discussion part of the manuscript and this reference (Gaym A, Pearson L, Soe KW. Maternity waiting homes in Ethiopia--three decades experience. Ethiopian medical journal, 2012 Jul;50(3):209-19) is reference number 13 currently and it was reference number 11 previously. But I can’t cite the result of the research, which is entitled by Newborn Outcomes and Maternity Waiting Homes in Low and Middle-Income Countries focusing on the impact of MWHs on neonatal health. 

5. Methods 

Comment:

a. Follow guideline for section and sub-sections.

Authors’ response: As per the guideline for section and sub-sections, we made revision.

5.1: P8, L – 170, make it as April 10, and delete /

Authors’ response: It is corrected as from March 10 to April 10, 2019

5.2: P8, L – 171, Authors should decide whether it should be as Gamo Gofa Zone or it should be used as Gamo Gofa zone in the whole manuscript including titles of tables and figures. Also authors should decide whether the percentages should be presented with one digit or two digit e.g. 3.2 or 3.21. These percentages be consist in the whole manuscript including tables and figures.

Authors’ response: we used Gamo Gofa zone in the whole document 

5.3: P8, L – 174 and 175, Instead of actual figures of population it can be written as about 160,000 and % male and female.

Authors’ response: The preposition about is included. 

5.4: P8, L – 190 – 194 either delete this or put it in supplementary information. This formula is not required.

Authors’ response: the formula is deleted 

5.5: P9, L – 198, Explain what is Kebele – International reader will not understand.

Authors’ response: We explained it as Kebeles (the smallest administrative units of Ethiopia)

5.6: P9, L – 203, Fig.2 - This figure can go into supplementary information.

Authors’ response: It is put in the supplementary information 

5.6: P9 – There is needed to link these two paragraphs giving information about data collection details. (L – 203 and L – 204).

Authors’ response: we create the link by using and variables such as

5.7: P9 – Line – 208. Write as insert Fig. 1 here.

Authors’ response: 

5.8: P10 – Line – 225. Write value of mean in the bracket. It should read – the mean () was to ----------

Authors’ response: mean (71.75%) is included 

5.9: P10 – Line – 234. Write value of mean in the bracket. It should read – the mean () was to ----------

Authors’ response: mean (31.28%) is included 

5.10: P10 – L 237 – Instead of scale and they – it should be scale and were

Authors’ response: It is corrected as scale, and were

5.11: P10 – L 247 – 248 – it should be --- health posts who were supervised by three supervisors (B.Sc, Nurses) from health centre.

Authors’ response: It is corrected as from health posts who were supervised by three supervisors (BSc, Nurses) from health centers were executed the data collection.

5.12: P11 – L 270 – Why this statement is required. Earlier you have not described this model. Or describe it little bit it here.

Authors’ response: We put this model because we checked the goodness of the final model using the value of Hosmer-Lemeshow model fitness test, and we decided that the final model is best fitted when the value is insignificant that is greater than 0.05. 

6. Results

Comment: Follow author’s guidelines for paragraph and sub-paragraphs font size, fond type, etc. Also decide number of decimals one or two to be presented percentages in the manuscript and Tables, and figures. It should be consistent.

Authors’ response: Revision is made based on the guideline 

6.1: P12 - L – 286 – Type – Insert table 1 here

Authors’ response: Table 1 is inserted 

6.2: P12 - L – 290 - Type – Insert table 2 here

Authors’ response: Table 2 is inserted 

6.3: P12 - L – 294 - Type – Insert table 3 here

Authors’ response: Table 3 is inserted 

6.4: P12 - L – 297 - Type – Insert Figure 3 here

Authors’ response: It is included in the supplementary information 

6.5: P12 - L – 303 - Type – Insert Figure 4 here

Authors’ response: It is inserted and labeled as Figure 2 because figure 2 and 3 included in the supplementary information 

Comment: Your result section description is short. In bivariate and multivariate section you bring AOR descriptions here from discussion section. In discussions do not put analysis figures. Give probable reasons, compare with other studies.

Authors’ response: we made a description of the bivariate and multivariate analysis in the result section 

7. Discussions

Comment: Discussion is quite long.

Authors’ response: We tried to minimize it. 

7.1: P14 L – 338 – Instead of Intention it should be intention.

Authors’ response: It is corrected as intention 

7.2: P15 L – 373 – 374 – It should be --- women who did not use it.

Authors’ response: It is corrected as women who did not use it 

7.3: P16 L – 391 – 392 – It should be --- (16,25,26).

Authors’ response: It is corrected as (16,25,26) 

7.4: P16 L – 397 – it should be – perceived

Authors’ response: It is corrected as perceived

7.5: P16 L – 414 – whether it is unlikely or likely. Check again.

Authors’ response: We checked it and it is unlikely 

7.6: P16 L – 415 – it should be – -----socio-demographic differences and availability of road and transportation between the populations.

Authors’ response: It is corrected as socio-demographic differences and availability of road and transportation between the populations. 

8. Recommendation

Comment: Follow the guidelines of journal. No separate recommendation section. It should be included in conclusions.

 Authors’ response: We corrected it based on the guidelines of the journal 

9. Declaration

9.1: P18 – L 451 – it should be sacrifice

Authors’ response: It is corrected as sacrifice 

9.2: P18 – L 458 – Debre Markos University collage of health sciences

Authors’ response: It is corrected as Debre Markos University College of Health Sciences 

10. References

Comment: Follow guidelines for references – how to write authors, and other details of article of the journal, book, chapter in book, website.

 Authors’ response: Dear reviewer based on your comments, we checked all references and we took correction for all. Reviewer #2: It is my pleasure to be designed as a reviewer of this paper, thank you very much!

This is a good paper that examined the intention to use MWH, one of the strategies to bridge geographic barriers to access obstetric care.

Line by line response to reviewer#2 comments 

Dear reviewer, we authors would like to thank you in advance for your valuable comments, and questions. And we replied for your concerns line by line. 

Comments

Background

1) The background section needs to be restructured for smooth flow from general problem to specific problem statement. In this case, please try to develop themes for each paragraph and link them logically to the issue. You may follow what is intention to use MWHs, what determines women’s intention to use MWHs (individual, community, health system factors) or what is known in the area, what is unknown, contribution of this paper to the scientific knowledge. You need also explain your theoretical framework here. Figure 1 is not described in the text

 Authors’ response: The background is restructured based on your comment, and also Figure 1 is described and cited. 

2) You use figures from 2016 EDHS data. Please update by 2019 mini-EDHS

Authors’ response: Based on your comment, we used mini EDHS 2019 report to indicate the current delivery service coverage but this report did not show the coverage of delivery services based on regions and also did not include maternal mortality rate. Thus we used both EDHS 2016, and mini 2019 reports. 

3) Page 5, paragraph 3 states transport challenges citing more than 8 years old data. Please update it. The extent problem regarding transport access is not the same as the problem before 2012. Over the last few years, the GoE distributed ambulances to districts to expand access to emergency transport though efficient use and universal access is still a challenge.

Authors’ response: Even if there is improvement, transport access is still a major problem, and still due to lack of modern transport, people use a locally made stretcher which is supported by the result of this study which revealed 72.24% of the respondents reported they were unlikely to get transportation access or walk to long distance in the study area. 

Methods

1. The study utilized pregnant women registered in the selected Kebeles as sampling frame and randomly select respondents from that frame. However, it is not well described how pregnant women were registered in the Kebele? How complete the registration was? And how the women or her household was located for interview? It is known the Ethiopian community health information system/family folder or pregnant women registration by HEWs/WDAs are incomplete. As such, this would induce selection bias due to incomplete sampling frame, potentially excluding eligible respondents

Authors’ response: In each kebele there are assigned HEWs and these HEWs provide community visits at house hold level at least 2 times a week and one of their responsibility is to investigate whether pregnant woman/s is present in that hose hold or not , and then register them. The registration includes the gestational age, EDD, Name, Phone number (if available), the exact location and others. Using this registration list of pregnant women as a frame work, we carried out simple random sampling technique, and interview was done at the household for those randomly selected pregnant women. No selection bias because it was random. To conduct community based study, no other best option to get pregnant women rather than the method that we used.

2. The analysis section doesn’t mention of account for clustering or not.

Authors’ response: This study was conducted in Gamo Gofa zone, and districts in this zone were considered as clusters because there is a homogeneous feature between districts. Due to the number of pregnant women estimation in each district, and our sample size, Kebma district was selected randomly. Then we considered that there is a heterogeneous feature with in a cluster, thus from 43 kebeles in Kemba district we took 25% of them (11 kebeles) randomly. Finally proportional allocation was done in each selected stratum (kebele) 

3. The outcome variable originally collected as ordinal but later reduced to binary for logistic regression analysis. I think this technique wastes information and may dilute the statistical power. Why you don’t use ordinal regression? 

Authors’ response: Yes, we agree with your comment, however many of the literatures that we cited provide binary logistic regression, and to make a discussion with these literatures, we have conducted binary logistic regression. 

4. Ethical clearance: Not clear whether written or verbal consent was obtained. And was ethical clearance or support letter this study sought from district health office?

Authors’ response: The consent that obtained from each participant was verbal consent and before we start data collection support letter was obtained from all respective bodies including the Kemba district health office. 

Results

1) Figure 2 and 3 are already narrated in the text. No need to present same information with different format. 

Authors’ response: I accept your comment, and both Figures are included in the supportive information 

2) Overall, the result section is presented multiple subheadings without adequate description of the findings. For instance, the Table 4 is narrated in the discussion section. I would bring the results described in the discussion section (narrations with figures with odds ratios) to here.

Authors’ response: Revision is made based on your comment 

3) Contrary to other studies, maternal education is not showing effect on the women’s intention to use MWHs. I think it can be related with the model fit or uncontrolled confounding. For instance, being government employee is associated with intention to use MWHs. This could be due to interaction or confounding with education. Please review it again.

Authors’ response: Model fitness was assured using Hosmer-Lemeshow goodness of fit and also the mode was built via forward stepwise method and we used Variance Inflation Factor (VIF) to check multicollinearity between independent variables, but no more than 15 % change of the B coefficient. 

Discussion

1. Summary the main findings in the first paragraph and dedicate subsequent paragraphs for discussing intention to use and major determinants. Don’t repeat results here (as described above please move the descriptions regarding Table 4 to results section

 Authors’ response: Revision is made based on your comment

2. There are potential biases in this study including selection bias described above, interviewer bias, and social desirability bias that are not mentioned in the discussion section.

Authors’ response: Revision is made based on your comment and selection bias, interviewer bias, and social desirability bias are mentioned in the discussion section as the following.

To avoid selection bias, we used simple random sampling technique to select the study participants from the list of pregnant women in the health extension workers registration book and the data were collected in community at home. During the interview, social desirability bias might be committed, but to reduce it only volunteer participants involved, and their names were kept anonymous, a brief overview of the study was provided, we used experienced interviewers, and the interview was carried out using a one-on-one strategy. To avoid interviewer bias, we instruct our interviewers to read each question accurately to the respondents to answer based on their best understanding of the question, do not interpret the question for the respondents, and offer to say again the question precisely as it appears. Generally, we educate our data collectors to avoid any adjustment, explanation, addition, subtraction, suggestion or change in verbal variation during the interview process.

3. Conclusion is simply repeating the main finding. Please rewrite to reflect the relevance of the findings to the program, implications to future research, or your concluding comments/take-home messages

Authors’ response: Revision is made based on your comment and it is corrected as the following 

 Generally, the intention to use maternity waiting home among pregnant women is less than half, indicated that it is not satisfactory which leads inaccessibility of obstetric cares and it will provoke pregnant women to give birth at home without skill birth attendants that may end up with development of different complications including maternal and neonatal deaths. And intention to use MWH is significantly associated with occupation of respondents, history of previous childbirth, past experiences of MWH utilization, direct and indirect subjective norm, and perceived behavioral control. These findings call for the intervention of the community to increase the intension of pregnant women towards MWH utilization and we authors recommend that:

4. Some of the recommendations are not grounded from the study findings. For instance, bullet # 2 on page 16

Authors’ response: we recommend bullet # 2 based on the findings of this study which revealed 81.82% of the respondents reported that it's difficult to get enough food in MWH during service utilization; and 72.24% of them reported they were unlikely to get transportation or walk to long distance.

5. Language

Finally, there are multiple grammatical errors that needs to edited by someone who has experiences in academic edition. For instance, page 4, paragraph 1 that reads..."one of the three..." not clear what are these; page 5, last paragraph first sentence talks delivery by SBA, but second sentence talks about home delivery. These are not parallel. The third sentence, "...within three.." need to be deleted; page 7, last paragraph first sentence,"... population proportion of 57.3%..." replace by population proportion of intention to use MWHs, 57.3%;

 Authors’ response: Based on your comment, language edition is made.

---

## [Decision Letter · Decision Letter 1]

29 Mar 2021

PONE-D-20-31704R1

Intention to use maternity waiting home and associated factors among pregnant women in Gamo Gofa zone, Southern Ethiopia, 2019

PLOS ONE

Dear Dr. Bitewa,

Thank you for submitting your manuscript to PLOS ONE. After careful consideration, we feel that it has merit but does not fully meet PLOS ONE’s publication criteria as it currently stands. Therefore, we invite you to submit a revised version of the manuscript that addresses the points raised during the review process.

We look forward to receiving your revised manuscript.

Kind regards,

Frank T. Spradley

Academic Editor

PLOS ONE

Reviewers' comments:

Reviewer's Responses to Questions

**Comments to the Author**

1. If the authors have adequately addressed your comments raised in a previous round of review and you feel that this manuscript is now acceptable for publication, you may indicate that here to bypass the “Comments to the Author” section, enter your conflict of interest statement in the “Confidential to Editor” section, and submit your "Accept" recommendation.

Reviewer #2: (No Response)

2. Is the manuscript technically sound, and do the data support the conclusions?

Reviewer #2: Yes

3. Has the statistical analysis been performed appropriately and rigorously? 

Reviewer #2: Yes

4. Have the authors made all data underlying the findings in their manuscript fully available?

Reviewer #2: No

5. Is the manuscript presented in an intelligible fashion and written in standard English?

Reviewer #2: No

6. Review Comments to the Author

Reviewer #2: This is my second review of this paper. Though it improved from the first version, still it has multiple flaws including grammatical errors and standard writing styles.

Major comments

1. 11 paragraphs introduction is too much for a paper. Reduce to 5-6 paragraphs.

2. Though you described how HEWs are updating pregnant women registration, that is not practical. You cannot be sure the registration is complete. In this case, I expect you to do ad hoc list of pregnant women with the help of the HEWs and WDAs before data collection. As such, this would induce selection bias due to incomplete sampling frame. You need to acknowledge this in the limitation section. Besides, you deployed HEWs as data collectors. This would also induce social desirability bias. Acknowledge this also as a limitation and discuss the implications of these biases on the observed results.

3. You used statistical criteria for selection of variables to include into the multivariate model. Why you did not use your theoretical framework? And you kept silent about the negative findings about the association between intention to use MWHs and the independent variables maternal education and attitude which is contrary to other studies. In such a behavioral study, it is good to base on conceptual frameworks to select variables. As such, you should force to retain “attitude” and “maternal education” in the model. It can also be related with the model fit or uncontrolled confounding. For instance, being government employee is associated with intention to use MWHs. This could be due to interaction or confounding with education. Besides, gravidity (previous history of childbirth) and past experience of MWHs would be correlated. Please review your analysis again, add possible interactions, force retain attitude and maternal education in the model. And if still, no significant associations with your outcome variable, please discuss the negative findings as well. These are important independent variables of interest.

4. Discussion section is still mere repetition of results (of course, you compared with previous literature). You did not summary the main findings, and interpret and discuss implications of your main findings. For instance, in your conclusion you mentioned subjective norm and perceived behavioral control as predictors to intention to use MWHs. You should interpret what subjective norms and perceived behavioral control mean and state as take home messages to readers. Think of it. Subjective norm and perceived behavioral control are still jargons. Can it be interpreted as low subjective norm or high community disapproval, low perceived benefits, and low self-efficacy. If so, you may summarize the main findings as such “The intention to use MWHs is low. Community disapproval, low self-efficacy, and maternal employment…are predictors to use MWHs” And discuss each main finding dedicating a paragraph, of course, without repeating the results.

5. Language: Still there are multiple grammatical errors that needs to edited by someone who has experiences in academic edition.

Minor comments

• Abstract: correct the # of respondents to 605

• Use of consistent decimal point across the text.

• You did not narrate the prevalence of intention to use MWHs in the results section. Please narrate it.

7. PLOS authors have the option to publish the peer review history of their article (what does this mean?). If published, this will include your full peer review and any attached files.

Reviewer #2: **Yes: **Gizachew Tadele Tiruneh

---

## [Author Response · Author response to Decision Letter 1]

16 Apr 2021

PONE-D-20-31704

Intention to use maternity waiting home and associated factors among pregnant women in Gamo Gofa zone, Southern Ethiopia, 2019

Dear Frank T. Spradley

Academic Editor of PLOS ONE journal 

We Authors would like to thank you for your constructive comments, and suggestions. In accordance with the reviewer’s comments, we made a revision. Therefore, we submitted a revised version of the manuscript that addresses the points raised during the review process.

Response to reviewers’ comments 

Dear Gizachew Tadele Tiruneh, we Authors would like to thank you for your helpful comments for the second time and based on the comments had given we made revision. 

Reviewer # 2: Major comments

1. 11 paragraphs introduction is too much for a paper. Reduce to 5-6 paragraphs.

Authors’ response: The introduction part of the manuscript is reduced to 5 paragraphs based on your comment.

2. Though you described how HEWs are updating pregnant women registration that is not practical. You cannot be sure the registration is complete. In this case, I expect you to do ad hoc list of pregnant women with the help of the HEWs and WDAs before data collection. As such, this would induce selection bias due to incomplete sampling frame. You need to acknowledge this in the limitation section. Besides, you deployed HEWs as data collectors. This would also induce social desirability bias. Acknowledge this also as a limitation and discuss the implications of these biases on the observed results.

Authors’ response: It was very difficult to conduct household survey to identify the number of pregnant women in that dispersed community. That is why we used the list of pregnant women from HEWs with its limitation (it may not include new pregnant women) to get the sampling frame. And acknowledgment is done for this limitation as your comment. The rationale of using HEWs as data collectors is that the sampling technique is simple random sampling and the secreted women in each kebele were very far apart to each other, and if we allocate another data collectors, it may bring false data. But the HEWs know the exact location of each pregnant woman and it is easy to collect the data which eliminate the probability of false recording. As you stated, using HEWs as data collectors might impose social desirability (response) bias and it might increase false positive response. However, the observed results did not show the presence of a significant response bias because the magnitude of intention to use MWH is less than 50% which is low, and the result of a favorable attitude, a favorable subjective norm and perceived behavioral control of the direct TPB components were below 50% and also the indirect attitude, indirect subjective norm, and indirect perceived behavioral control of the respondents were about 50% and less than it and about 50% of the participants agreed that HEWs thinks that they should stay in MWH and about 50% the respondents also reported that HEWs’ approval for their stay in MWH was much important. Similarly, more than 80% of the respondents reported that it's unlikely to get enough food in MWH. So based on these and other findings, we believed that the episode of social desirability bias was very minimal/unlikely. Even if it is, we acknowledged it in the limitation part based on your comment. 

3. You used statistical criteria for selection of variables to include into the multivariate model. Why you did not use your theoretical framework? And you kept silent about the negative findings about the association between intention to use MWHs and the independent variables maternal education and attitude which is contrary to other studies. In such a behavioral study, it is good to base on conceptual frameworks to select variables. As such, you should force to retain “attitude” and “maternal education” in the model. It can also be related with the model fit or uncontrolled confounding. For instance, being government employee is associated with intention to use MWHs. This could be due to interaction or confounding with education. Besides, gravidity (previous history of childbirth) and past experience of MWHs would be correlated. Please review your analysis again, add possible interactions, force retain attitude and maternal education in the model. And if still, there is no significant associations with your outcome variable, please discuss the negative findings as well. These are important independent variables of interest.

Authors’ response: To select variables for the final model we considered two methods: 1) the statistical method that is P-value of less than 0.25 rather than 0.05 to increase the number of variables, which is one of the recommended methods for variable selection. 2) If the variable is significantly associated with intention to use MWH in other researches and if that variable is very important predictor of the outcome, we considered it to be included in the finale model. We prefer to use the statistical method of variable selection rather than the theoretical framework because the final model result is very disrupted if we include a variable which is extremely insignificant in the bivariate analysis and also which is not significant in other research findings. But fortunately, the variables maternal education and both direct and indirect attitudes of the respondents were included in the final model because the p-value of these variables was less than 0.25 and then became insignificant in the multivariate analysis. For your evidence, in the bivariate analysis age of the respondents, educational level, occupation, previous childbirth history, experience of MWH utilization, reason for past utilization, and duration of stay at MWH, direct and indirect attitude, direct and indirect subjective norm, and direct and indirect perceived behavioral control were associated with intention to use MWH with P-value of less than 0.25. To assess the interaction between maternal educational level and government employee, and between previous childbirth history and past experience of MWH utilization, we created new variables and these two new variables were included in the model but were remain insignificant with p-value of greater than 0.05. Based on your comment the negative findings also discussed. 

4. Discussion section is still mere repetition of results (of course, you compared with previous literature). You did not summary the main findings, and interpret and discuss implications of your main findings. For instance, in your conclusion you mentioned subjective norm and perceived behavioral control as predictors to intention to use MWHs. You should interpret what subjective norms and perceived behavioral control mean and state as take home messages to readers. Think of it. Subjective norm and perceived behavioral control are still jargons. Can it be interpreted as low subjective norm or high community disapproval, low perceived benefits, and low self-efficacy? If so, you may summarize the main findings as such “The intention to use MWHs is low. Community disapproval, low self-efficacy, and maternal employment…are predictors to use MWHs” And discuss each main finding dedicating a paragraph, of course, without repeating the results.

Authors’ response: Discussion part of the manuscript is revised based on your comments. In this study subjective norm of pregnant women to use MWHs is measured as favorable or unfavorable based on the mean value which indicated that pregnant women with unfavorable subjective norm have low subjective norm or high community disapproval. And the concept of self-efficacy is used as perceived behavioral control, which means the perception of the ease or difficulty of pregnant women to use MWHs and unfavorable perceived behavioral control indicates low self-efficacy. 

5. Language: Still there are multiple grammatical errors that need to edit by someone who has experiences in academic edition.

Authors’ response: Basic edition is carried out in the whole document.

Minor comments

• Abstract: correct the # of respondents to 605: It is corrected from 621 to 605

• Use of consistent decimal point across the text. It is checked. 

• You did not narrate the prevalence of intention to use MWHs in the results section. Please narrate it. It is already narrated in the first 4 lines of the result section of the manuscript without subheadings but now subheading “prevalence of intention to use MHHs” is included.

---

## [Editor Report · Decision Letter 2]

22 Apr 2021

Intention to use maternity waiting home and associated factors among pregnant women in Gamo Gofa zone, Southern Ethiopia, 2019

PONE-D-20-31704R2

Dear Dr. Bitewa,

We’re pleased to inform you that your manuscript has been judged scientifically suitable for publication and will be formally accepted for publication once it meets all outstanding technical requirements.

Kind regards,

Frank T. Spradley

Academic Editor

PLOS ONE

---

## [Editor Report · Acceptance letter]

29 Apr 2021

PONE-D-20-31704R2 

Intention to use maternity waiting home and associated factors among pregnant women in Gamo Gofa zone, Southern Ethiopia, 2019  

Dear Dr. Bitewa:

I'm pleased to inform you that your manuscript has been deemed suitable for publication in PLOS ONE. Congratulations! Your manuscript is now with our production department. 

Kind regards, 

on behalf of

Dr. Frank T. Spradley 

Academic Editor

PLOS ONE